# Molecular Russian dolls

Kang Cai [1], Mark C. Lipke[1], Zhichang Liu[2], Jordan Nelson[1,3], Tao Cheng [4], Yi Shi[1], Chuyang Cheng[1], Dengke Shen[1], Ji-Min Han[1,7], Suneal Vemuri[1], Yuanning Feng[1], Charlotte L. Stern[1], William A. Goddard III [4], Michael R. Wasielewski[1,3] & J. Fraser Stoddart [1,5,6]

The host-guest recognition between two macrocycles to form hierarchical non-intertwined ring-in-ring assemblies remains an interesting and challenging target in noncovalent synthesis. Herein, we report the design and characterization of a box-in-box assembly on the basis of host-guest radical-pairing interactions between two rigid diradical dicationic cyclophanes. One striking feature of the box-in-box complex is its ability to host various 1,4-disubstituted benzene derivatives inside as a third component in the cavity of the smaller of the two diradical dicationic cyclophanes to produce hierarchical Russian doll like assemblies. These results highlight the utility of matching the dimensions of two different cyclophanes as an efficient approach for developing new hybrid supramolecular assemblies with radical-paired ring-in-ring complexes and smaller neutral guest molecules.

[1] Department of Chemistry, Northwestern University, 2145 Sheridan Road, Evanston, IL 60208, USA. [2] School of Science, Westlake University, 18, Shilongshan Road, 310024 Hangzhou, China. [3] Institute for Sustainability and Energy at Northwestern, Northwestern University, 2145 Sheridan Road, Evanston, IL 60208, USA. [4] Materials and Process Simulation Center, California Institute of Technology (MC139-74), Pasadena, CA 91125, USA. [5] Institute for Molecular Design and Synthesis, Tianjin University, Nankai District, 300092 Tianjin, People's Republic of China. [6] School of Chemistry, University of New South Wales, Sydney, NSW 2052, Australia. [7] Present address: State Key Laboratory of Explosion Science and Technology of China, Beijing Institute of Technology, 5 South Zhongguancun Street, 100081 Beijing, China. Correspondence and requests for materials should be addressed to J.F.S. (email: stoddart@northwestern.edu)

The past five decades have witnessed the rise and flourishing of host–guest and supramolecular chemistry[1–12]. Various macrocyclic hosts have been described in the literature based on diverse noncovalent bonding interactions for recognizing specific guests such as small molecules[13], ions[14], biomolecules[15], fullerenes[16], and more[17]. The host–guest recognition between two macrocycles to form hierarchical non-intertwined ring-in-ring or host-in-host assemblies, however, remains an interesting and challenging target in noncovalent synthesis[18–25]. Such complexes are promising precursors for constructing Russian doll-like superstructures[26–31] or higher order mechanically interlocked molecules, such as molecular Borromean rings[32,33].

Tetracationic cyclophanes, formed by linking together two $\pi$-electron-deficient dicationic 4,4′-bipyridinium ($BIPY^{2+}$) units, represent an important class of synthetic hosts on account of their ability to bind electron-rich guests to form 1:1 or 1:2 host–guest complexes[34]. Cyclobis(paraquat-p-phenylene)[35] ($CBPQT^{4+}$) is the most intensively explored (Fig. 1) among these tetracationic cyclophanes because the 6.8 Å centroid-to-centroid distance[13] between the two $BIPY^{2+}$ units of this host which is well-suited for accommodating electron-rich planar aromatic guests such as tetrathiafulvalene[36] (TTF), 1,4-bis[2-(2-hydroxyethoxy)]ethoxylbenzene[37], and others[38]. The diradical dicationic form—$CBPQT^{2(+\bullet)}$—of $CBPQT^{4+}$ can also act as a host, encapsulating 1,1′-dialkyl-4,4′-bipyridinium radical cations ($BIPY^{+\bullet}$) to form a trisradical tricationic complex $[BIPY \subset CBPQT]^{3(+\bullet)}$. The binding in these trisradical complexes is relatively strong as a result of homophilic radical-pairing interactions (Fig. 1) which represented the first examples of radical–radical interactions between host and guest pairs[39]. Many mechanically interlocked molecules[40,41] and redox-switchable molecular machines[42,43] have been developed subsequently by employing $[BIPY \subset CBPQT]^{3(+\bullet)}$ as a recognition motif.

Recently, we have expanded[44] the radical recognition motif to a tetraradical tetracationic host–guest complex in which a square-shaped diradical cyclophane [cyclobis(paraquat-4,4′-biphenylene)]$^{2(+\bullet)}$ ($SqBox^{2(+\bullet)}$) encapsulates a cyclobis(paraquat-m-phenylene) ($m$-$CBPQT^{2(+\bullet)}$) guest. Recognition between these two diradical dicationic cyclophanes is highly selective, while $CBPQT^{2(+\bullet)}$ and $SqBox^{2(+\bullet)}$ exhibit[44,45] no binding affinity as a consequence of their mismatched sizes. The $m$-$CBPQT^{2(+\bullet)}$ cyclophane is, however, unable to encapsulate any guests on account of the relatively small spacing distance between two

$BIPY^{+\bullet}$ units, thus precluding its use in the formation of more sophisticated hierarchical superstructures.

Herein, we report the design and synthesis of a rigid tetracationic cyclophane $1^{4+}$ with a centroid-to-centroid distance (Fig. 1) between two $BIPY^{2+}$ units of 13.1 Å, which, in its diradical dicationic state $1^{2(+\bullet)}$, is ideal for encapsulating $CBPQT^{2(+\bullet)}$. The strong radical-pairing interactions between $1^{2(+\bullet)}$ and $CBPQT^{2(+\bullet)}$ has led to the formation of a unique tetraradical tetracationic, box-in-box complex. Interestingly, this box-in-box complex can accommodate small aromatic guest molecules inside the void of the $CBPQT^{2(+\bullet)}$ component to generate a series of tetraradical tetracationic Russian doll assemblies, both in solution and in the solid state. This kind of complexation is rare in the case of organic compounds[26–31].

## Results

### Synthesis and structural characterization of 1·4PF$_6$.
The tetracationic cyclophane $1\cdot4PF_6$ was obtained (Fig. 2) in 42% yield by the 1:1 cyclization between $2$ and $3\cdot2PF_6$ with tetrabutylammonium iodide (TBAI) as a catalyst. $1\cdot4PF_6$ was characterized by $^1H$, $^{13}C$, $^1H$–$^1H$ COSY and NOE NMR spectroscopies (Supplementary Figures 1-4), all of which are in support of the existence of a pure highly symmetrical species in solution. Additional evidence for the formation of $1\cdot4PF_6$ was obtained by high-resolution mass spectrometry (HRMS), which detected the species $[M - PF_6]^+$ and $[M - 2PF_6]^{2+}$ in the gas phase at $m/z = 1152.2163$ and 505.1264, respectively. Single crystals were obtained by slow vapor diffusion of $^iPr_2O$ into a MeCN solution of $1\cdot4PF_6$ during 4 days. The solid-state structure of $1\cdot4PF_6$, which was determined by single-crystal X-ray diffraction (XRD) analysis, indicates (Fig. 2 and Supplementary Figure 15) that the distance between the two $BIPY^{2+}$ units is ca. 13.1 Å. This separation is expected to be ideal for recognition of $CBPQT^{2(+\bullet)}$, considering that $CBPQT^{2(+\bullet)}$ features[46] a 6.9 Å separation of its $BIPY^{+\bullet}$ units, and radical–radical interactions between $BIPY^{+\bullet}$ units typically have spacings[46,47] of 3.1–3.3 Å.

### Solution phase characterization of the box-in-box complex.
The association between $1^{2(+\bullet)}$ and $CBPQT^{2(+\bullet)}$ was investigated in the first instance by UV–Vis–NIR spectroscopy. Solutions of $1\cdot4PF_6$ (0.5 mM) and $CBPQT\cdot4PF_6$ (0.5 mM) in MeCN were reduced by Zn dust to generate the corresponding radical

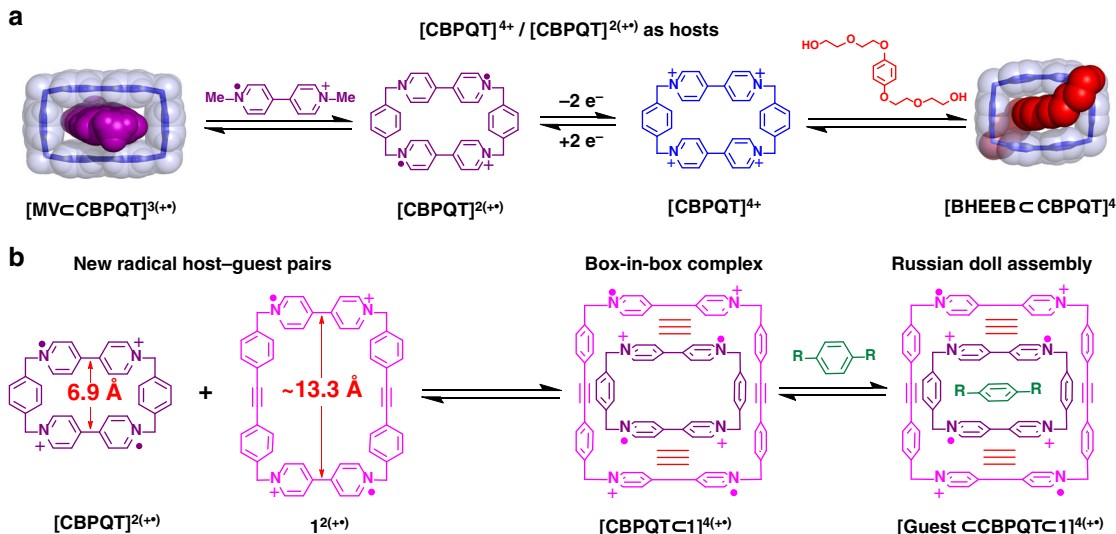

**Fig. 1** Design and structures. **a** Examples of $CBPQT^{4+}$ and $CBPQT^{2(+\bullet)}$ as hosts. **b** Proposed superstructures of the tetraradical tetracationic box-in-box complex and Russian doll assemblies

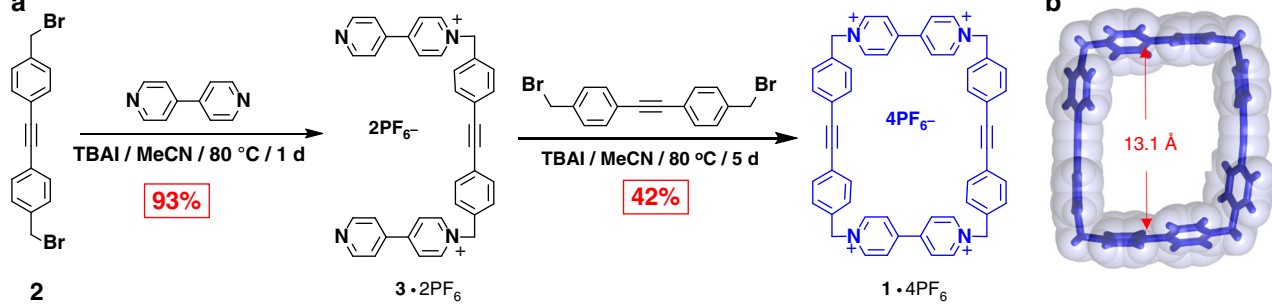

**Fig. 2** Synthesis and crystal structure. **a** Synthesis of **1**·4PF₆ and **b** solid-state structure of **1**⁴⁺

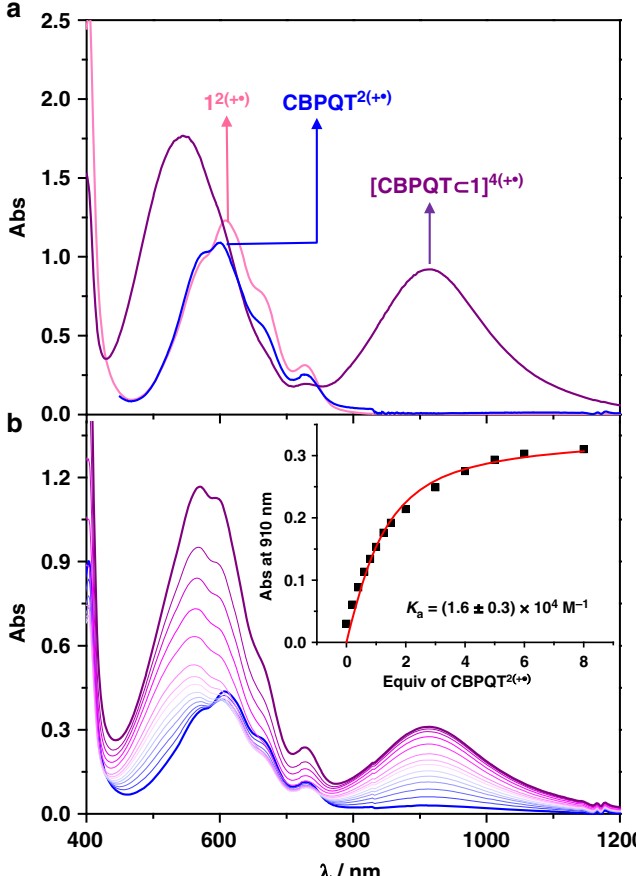

**Fig. 3** UV–Vis–NIR characterization of [**CBPQT ⊂ 1**]⁴⁽⁺•⁾. **a** UV–Vis–NIR spectra (0.50 mM in MeCN, 1 mm path cuvette) of **CBPQT**²⁽⁺•⁾ (blue), **1**²⁽⁺•⁾ (red), and a 1:1 molar ratio of **CBPQT**²⁽⁺•⁾ and **1**²⁽⁺•⁾ (purple); **b** Vis/NIR Spectra (MeCN, 2 mm cuvette) on titrating **CBPQT**²⁽⁺•⁾ into **1**²⁽•⁺⁾ (0.10 mM). Initial and final spectra are highlighted in blue and purple, respectively. The inset shows the change in absorption at 910 nm on titration of **1**²⁽⁺•⁾ with **CBPQT**²⁽⁺•⁾. Curve fitting is highlighted in red

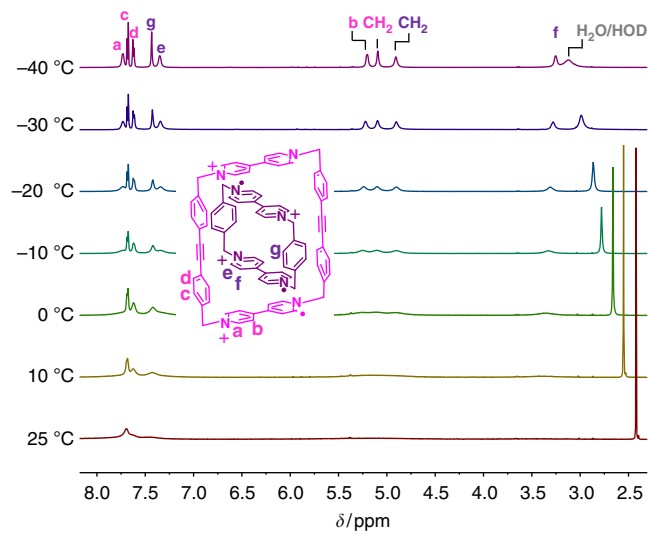

**Fig. 4** VT-NMR studies of [**CBPQT ⊂ 1**]⁴⁽⁺•⁾. ¹H NMR Spectra recorded from −40 to +25 °C for a 1:1 molar mixture of **CBPQT**²⁽⁺•⁾ and **1**²⁽•⁺⁾ (1.0 mM each) in CD₃CN

cyclophanes **1**²⁽⁺•⁾ and **CBPQT**²⁽⁺•⁾, respectively. Both **1**²⁽⁺•⁾ and **CBPQT**²⁽⁺•⁾ display (Fig. 3a) very similar absorption bands around 600 nm, an observation which is characteristic of radical cationic BIPY⁺• absorptions in the absence of radical-pairing interactions. The spectrum of a 1:1 mixture of **1**²⁽⁺•⁾ and **CBPQT**²⁽⁺•⁾, however, displays an intense new NIR absorption band around 910 nm characteristic[44,48] of radical cationic (BIPY⁺•)₂ dimers. This observation indicates the formation of a complex between **1**²⁽⁺•⁾ and **CBPQT**²⁽⁺•⁾ as a result of strong radical–radical interactions between the BIPY⁺• units of **1**²⁽⁺•⁾

and **CBPQT**²⁽⁺•⁾. In order to estimate the binding affinity between **1**²⁽⁺•⁾ and **CBPQT**²⁽⁺•⁾, UV–Vis–NIR absorption titrations were carried out by monitoring the change of the NIR absorption band centered on 910 nm. The resulting data could be made to fit a 1:1 isotherm to give (Fig. 3b) a binding constant of $K_a = (1.6 \pm 0.3) \times 10^4 \, M^{-1}$, which is comparable with that ($K_a = 20,000–30,000 \, M^{-1}$ for $N$-alkyl-substituted BIPY⁺•) of [**BIPY ⊂ CBPQT**]³⁽⁺•⁾[46,47], and smaller[44] than that ($K_a = 1.12 \pm 0.08 \times 10^5 \, M^{-1}$) of [$m$-**CBPQT ⊂ SqBox**]⁴⁽⁺•⁾.

Since the viologen diradical dicationic dimer (BIPY⁺•)₂ and the tetraradical tetracationic complex [$m$-**CBPQT ⊂ SqBox**]⁴⁽⁺•⁾ are reported[44,49] to be diamagnetic, based on EPR measurements, the tetraradical tetracationic host–guest complex [**CBPQT ⊂ 1**]⁴⁽⁺•⁾ was also expected to be diamagnetic. Thus, we sought to characterize the host–guest complex by ¹H NMR spectroscopy, as we have previously succeeded[45] in doing for the tetraradical tetracationic state of a rotaxane, based on the [$m$-**CBPQT ⊂ SqBox**]⁴⁽⁺•⁾ recognition motif. At room temperature, however, only a single broad peak near 7.7 ppm was observed in the spectrum (Fig. 4) of a 1:1 mixture of **1**²⁽⁺•⁾ and **CBPQT**²⁽⁺•⁾ in CD₃CN, presumably because of a small thermal population of a paramagnetic state at room temperature[45,49]. Upon cooling to −20 °C, the signals of the complex are individually resolved, yet remain broad until further cooling to −40 °C, at which point all of the resonances of the host–guest complex are displayed (Fig. 4) as sharp signals.

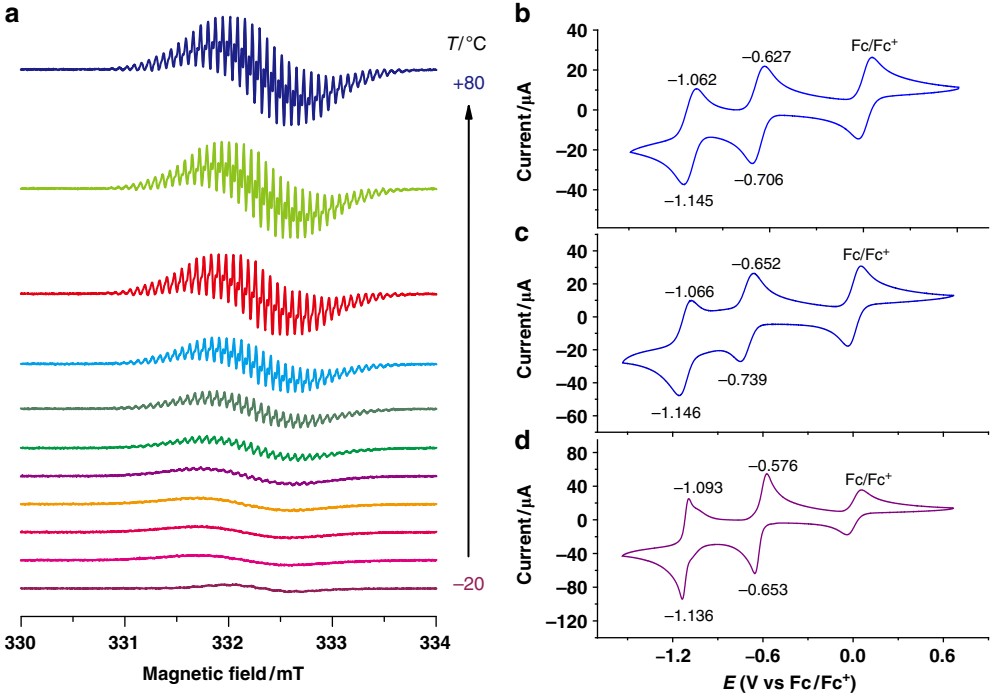

**Fig. 5** EPR characterization. **a** EPR Spectra recorded in 10 degree interval on a 1:1 molar mixture of **CBPQT**$^{2(+\bullet)}$ and **1**$^{2(+\bullet)}$ (0.50 mM each) in MeCN. Cyclic voltammograms of **b** **1**$^{4+}$ (0.20 mM), **c** **CBPQT**$^{4+}$ (0.20 mM), and **d** a 1:1 molar mixture of **1**$^{4+}$ and **CBPQT**$^{4+}$ (0.20 mM each)

Since [**CBPQT** ⊂ **1**]$^{4(+\bullet)}$ is the first radical-pairing-based host–guest complex to be characterized by $^1$H NMR spectroscopy, we sought to probe the limits of this technique more deeply for studying supramolecular assemblies featuring radical–radical interactions. Thus, $^1$H–$^1$H COSY NMR spectra were recorded (Supplementary Figure 6) at –40 °C in order to assign the peaks of the host–guest complex. Noticeably, both signals for the aromatic and CH$_2$ protons on **CBPQT**$^{2(+\bullet)}$ are shifted significantly to higher field (Fig. 4) as a consequence of the stronger shielding effect of the host–guest interactions compared with that of **1**$^{2(+\bullet)}$. Resonances for the *meta* positions of the pyridinium units in **1**$^{2(+\bullet)}$ and **CBPQT**$^{2(+\bullet)}$ which were located at 5.22 (peak b in Fig. 4) and 3.28 ppm (peak f in Fig. 4), respectively, are shifted to considerably higher fields than those of the *ortho* protons of the pyridinium units (7.73 ppm for peak a and 7.34 ppm for peak e in Fig. 4). This observation is in line with the solid-state superstructure (Fig. 6, see below) of the complex [**CBPQT** ⊂ **1**]$^{4(+\bullet)}$, which shows that the BIPY$^{+\bullet}$ units in the radical dimers (BIPY$^{+\bullet}$)$_2$ are nearly perpendicular to each other, torsion angle 84°, such that the *meta* CHs of the BIPY$^{+\bullet}$ units in **1**$^{2(+\bullet)}$ and **CBPQT**$^{2(+\bullet)}$ are strongly shielded[45], while the *ortho* CHs are not shielded.

CW-EPR spectroscopy experiments were carried out in order to probe the electronic structure of [**CBPQT** ⊂ **1**]$^{4(+\bullet)}$. The EPR spectrum of **CBPQT**$^{2(+\bullet)}$ (298 K, 0.50 mM, Supplementary Figure 22) displays almost no hyperfine structure. This observation is a characteristic result of the unpaired electrons in the two BIPY$^{+\bullet}$ units being close enough together for electronic overlap such that spin-exchange interactions between them dominate the spectrum[39]. The EPR (298 K, 0.50 mM, Supplementary Figure 22) spectrum of **1**$^{2(+\bullet)}$ exhibits, however, hyperfine splitting which resembles that of other *N,N*′-dialkyl substituted BIPY$^{+\bullet}$ radical cations reported in the literature[39,46], since the relatively longer separation distance between two BIPY$^{+\bullet}$ units leads to the absence of efficient overlap for spin-exchange interactions. In comparison, the EPR spectrum (298 K, 0.25 mM each, Supplementary Figure 22) of a 1:1 mixture of **1**$^{2(+\bullet)}$ and **CBPQT**$^{2(+\bullet)}$

exhibits relatively minimal hyperfine structure. This implies a disruption of the spin-exchange interactions between the two BIPY$^{+\bullet}$ radicals by an interaction (complexation) between **1**$^{2(+\bullet)}$ and **CBPQT**$^{2(+\bullet)}$. Variable-temperature EPR spectra (Fig. 5a) of the 1:1 mixture of **1**$^{2(+\bullet)}$ and **CBPQT**$^{2(+\bullet)}$ in MeCN was then measured at temperatures ranging from –20 to 80 °C. The EPR spectra at −20 °C of the 1:1 mixture displays almost no microwave absorption by unpaired radicals. This indicates that there is strong association of the cyclophanes to form the radical-paired ground state of the complex [**CBPQT** ⊂ **1**]$^{4(+\bullet)}$, an observation which is consistent with the VT-NMR results. Upon heating from −20 to 80 °C, the EPR signal of the mixture gradually increased in intensity, along with an increase in the appearance of hyperfine structure. This observation is ascribed to the temperature-induced dissociation of the box-in-box complex into its individual radical cyclophane components.

In order to gain insight into the redox processes involved in the assembly and disassembly of [**CBPQT** ⊂ **1**]$^{4(+\bullet)}$, cyclic voltammetry (CV) experiments were performed using ferrocene as the internal redox standard. Both **CBPQT**·4PF$_6$ and **1**·4PF$_6$ (concentration: 0.20 mM in MeCN containing 0.1 M NBu$_4$PF$_6$, scan rate: 0.2 V/s) exhibit (Fig. 5b, c) similar reversible redox waves, though the waves for **CBPQT**·4PF$_6$ are more positively shifted than those of **1**·4PF$_6$, presumably because of its higher ring strain, resulting from the smaller ring size. CVs of an equimolar mixture of **CBPQT**·4PF$_6$ and **1**·4PF$_6$ were also measured (Fig. 5d) at a concentration of 0.20 mM for each box. With a scan rate of 0.2 V/s, the equimolar mixture displays two sharp reversible redox waves, similar to those of the individual boxes, but shifted to more positive potentials. The shifts in the potentials of the redox waves for the mixture of **CBPQT**·4PF$_6$ and **1**·4PF$_6$ can be rationalized by the formation of a tetraradical tetracationic host–guest complex since radical-pairing interactions will lend stability to the radical cationic redox states, as observed previously[39,46] for other radical-based host–guest complexes. Furthermore, CV experiments which were carried out at different scan rates show scan rate-dependent CV behavior

(Supplementary Figure 14) similar[46] to that of the [**BIPY** ⊂ **CBPQT**]$^{3(+\bullet)}$ complex.

**Solid-state structure of the box-in-box complex.** Single crystals of [**CBPQT** ⊂ **1**]•4PF$_6$ were obtained by vapor diffusion of $^i$Pr$_2$O at room temperature under an atmosphere of N$_2$ into a solution of an equimolar mixture of **1**•4PF$_6$ and **CBPQT**•4PF$_6$ which had been reduced over Zn dust in MeCN and filtered. Single-crystal X-ray diffraction analysis revealed (Fig. 6, Supplementary Figure 16) a solid-state superstructure in which **CBPQT**$^{2(+\bullet)}$ is encapsulated inside the cavity of the **1**$^{2(+\bullet)}$ ring to form a 1:1 host–guest inclusion complex that is consistent with the results found in solution.

The **CBPQT**$^{2(+\bullet)}$ ring in the [**CBPQT** ⊂ **1**]$^{4(+\bullet)}$ complex has a larger BIPY$^{+\bullet}$ centroid-to-centroid distance (7.19 Å) than that[49] observed (6.92 Å) in the solid-state structure of **CBPQT**•2PF$_6$. The BIPY$^{+\bullet}$ centroid-to-centroid distance is 13.2 Å for **1**$^{2(+\bullet)}$ in the [**CBPQT** ⊂ **1**]$^{4(+\bullet)}$ complex (Fig. 6c), which is similar to that (13.1 Å) obtained for **1**•4PF$_6$. This observation indicates that the **CBPQT**$^{2(+\bullet)}$ ring has to expand slightly in order to optimize binding of **1**$^{2(+\bullet)}$ in its cavity.

The torsional angle between BIPY$^{(+\bullet)}$ units of **1**$^{2(\bullet+)}$ and **CBPQT**$^{2(+\bullet)}$ is about 84° (Fig. 6d), which is an energetically favorable angle for BIPY$^{\bullet+}$ radical pimers, according to theoretical predictions[48]. It is noteworthy that the centroid-to-centroid distance between the adjacent viologen units of **1**$^{2(+\bullet)}$ and **CBPQT**$^{2(+\bullet)}$ is only 3.03 Å (Fig. 6a, b), which is shorter than the spacings (3.1–3.3 Å) that are usually observed[44,46,47] for viologen radical-pairing interactions. This distance indicates the presence of strong radical-pairing interactions in the solid state, an observation which is consistent with the results of UV–Vis–NIR and $^1$H NMR spectroscopic investigations in solution.

The extended superstructure reveals (Supplementary Figure 16d) intermolecular co-facial π-stacking between the radical

cationic viologen units of **1**$^{2(+\bullet)}$, which are similar to that observed[46,47] in the solid-state superstructure of [**BIPY** ⊂ **CBPQT**]$^{3(+\bullet)}$. The centroid-to-centroid separation is 3.48 Å between the co-facially stacked BIPY$^{+\bullet}$ radical cationic subunits of adjacent [**CBPQT** ⊂ **1**]$^{4(+\bullet)}$ complexes, a distance that is typical for [π···π] interactions, but longer than those (3.1–3.3 Å) typically observed[44,46,47] for BIPY$^{+\bullet}$ radical-pairing. This observation suggests that the strength of radical-pairing interactions between adjacent complexes might be attenuated since all of the single electrons of the **BIPY**$^{+\bullet}$ units of **1**$^{2(\bullet+)}$ and **CBPQT**$^{2(+\bullet)}$ are strongly paired in interactions that are localized within the [**CBPQT** ⊂ **1**]$^{4(+\bullet)}$ complex.

**Formation of Russian doll assembly.** Notably, the solid-state superstructure of the box-in-box complex [**CBPQT** ⊂ **1**]$^{4(+\bullet)}$, confirmed the presence of a void inside of the smaller **CBPQT**$^{2(+\bullet)}$ cyclophane component that is large enough to potentially accommodate a second guest. We thus sought out guest molecules that could occupy this cavity to form a Russian doll-like assembly. Radical cationic dimethyl viologen (**MV**$^{+\bullet}$) was firstly employed as the guest in order to investigate the possibility of forming a pentaradical pentacationic [**MV** ⊂ **CBPQT** ⊂ **1**]$^{5(+\bullet)}$ complex. Neither UV–Vis–NIR spectroscopy nor X-ray crystallography, however, indicated the formation of this three-component assembly. We attribute this observation to the limited space between the $p$-phenylene and bis($p$-phenylene)ethyne spacers of the two cyclophanes in [**CBPQT** ⊂ **1**]$^{4(+\bullet)}$, which might not provide a large enough window for **MV**$^{+\bullet}$ to thread.

Smaller neutral guest molecules, such as 1,4-dichlorobenzene, were employed in an effort to find guests that could fit into the cavity of the [**CBPQT** ⊂ **1**]$^{4(+\bullet)}$ complex. These investigations were facilitated by the ability to observe well-defined $^1$H NMR spectra of [**CBPQT** ⊂ **1**]$^{4(+\bullet)}$ at reduced temperatures, and thus, 1 equiv 1,4-dichlorobenzene was added to a 1.0 mM solution of [**CBPQT** ⊂ **1**]$^{4(+\bullet)}$ in CD$_3$CN, and VT $^1$H NMR spectra were recorded. Upon decreasing the temperature from 25 to −40 °C, the signals (Fig. 7) of a three-component [$p$-C$_6$H$_4$Cl$_2$ ⊂ **CBPQT** ⊂ **1**]$^{4(+\bullet)}$ complex gradually appeared and sharpened into well-resolved signals. The formation of the three-component complex was evident from the change in the chemical shifts of all the peaks of the two cyclophane components from those that

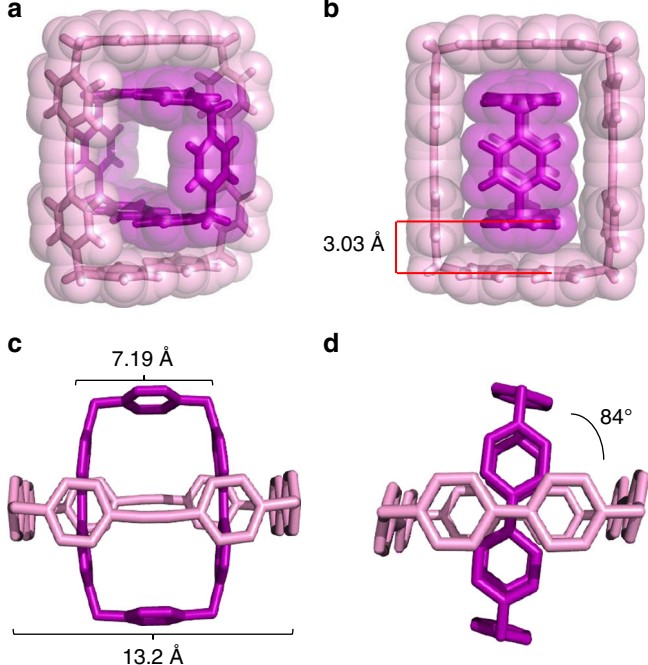

**Fig. 6** Solid-state superstructures of [**CBPQT** ⊂ **1**]$^{4(+\bullet)}$. **a** Perspective and **b** plan views depicted as tubular and space-filling representations. **c**, **d** Different side-on views, depicted as tubular representations. Hydrogen atoms in **c** and **d** are omitted for the sake of clarity

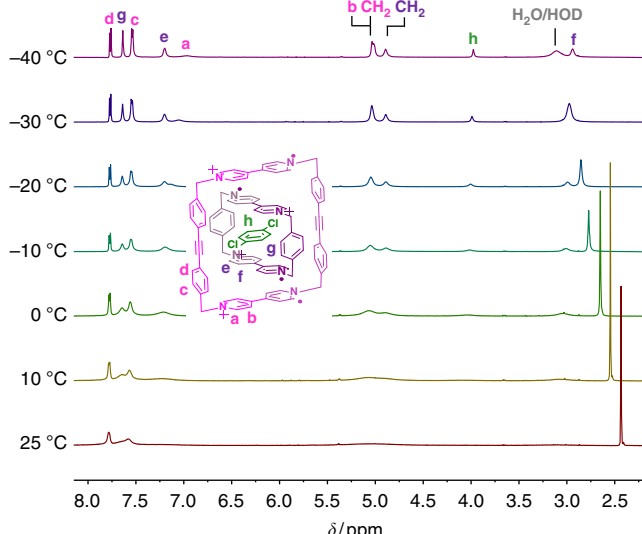

**Fig. 7** VT-NMR studies of [$p$-C$_6$H$_4$Cl$_2$ ⊂ **CBPQT** ⊂ **1**]$^{4(+\bullet)}$ $^1$H NMR Spectra recorded from −40 to +25 °C for a 1:1:1 molar mixture of **CBPQT**$^{2(+\bullet)}$, **1**$^{2(+\bullet)}$ and 1,4-dichlorobenzene (1.0 mM each) in CD$_3$CN

were observed (Fig. 4) for the two component complex $[\mathbf{CBPQT} \subset \mathbf{1}]^{4(+\bullet)}$ in the absence of 1,4-dichlorobenzene. The signals of the BIPY$^{+\bullet}$ units, in particular, were shifted significantly to higher fields. The encapsulating of 1,4-dichlorobenzene was further confirmed by the observation (Fig. 7) of a new singlet resonance for this guest near 4 ppm. All these observations indicated that 1,4-dichlorobenzene was encapsulated by $[\mathbf{CBPQT} \subset \mathbf{1}]^{4(+\bullet)}$ in CD$_3$CN solution, leading to upfield chemical shifts of the signals of the cyclophane components of the complex and to ca. 4 ppm for the 1,4-dichlorobenzene guest that is surrounded on both sides by the $\mathbf{CBPQT}^{2(+\bullet)}$ component of the three-component $[\mathit{p}\text{-}\mathbf{C_6H_4Cl_2} \subset \mathbf{CBPQT} \subset \mathbf{1}]^{4(+\bullet)}$ complex.

One matter that is worth investigating is whether the inclusion of the third component guest ($\mathit{p}$-C$_6$H$_4$Cl$_2$) will affect the binding affinity between $\mathbf{CBPQT}^{2(+\bullet)}$ and $\mathbf{1}^{2(+\bullet)}$. Thus, we repeated the UV–Vis–NIR titration experiments with $\mathit{p}$-C$_6$H$_4$Cl$_2$ presented in the solution. By titrating $\mathbf{CBPQT}^{2(+\bullet)}$ into the solution mixture of $\mathbf{1}^{2(+\bullet)}$ and a number of equivalents of $\mathit{p}$-C$_6$H$_4$Cl$_2$, similar 1:1 fitting isotherms can be applied to the titration data, and "apparent" binding constants were obtained. Notably, the existence of $\mathit{p}$-C$_6$H$_4$Cl$_2$ guests did, indeed, increase the binding affinity between $\mathbf{1}^{2(+\bullet)}$ and $\mathbf{CBPQT}^{2(+\bullet)}$; the more equivalents of $\mathit{p}$-C$_6$H$_4$Cl$_2$ that were added, the more the binding affinity increased. For instance, when 2.0 eq. $\mathit{p}$-C$_6$H$_4$Cl$_2$ was present, the "apparent" binding constant was found to be $(3.8 \pm 0.3) \times 10^4$ M$^{-1}$ (Supplementary Figure 11), which is 1.3 times higher than that of the box-in-box complex without $\mathit{p}$-C$_6$H$_4$Cl$_2$; and when the equivalents of $\mathit{p}$-C$_6$H$_4$Cl$_2$ were 6.7 and 20, the "apparent" binding constant increased to $(6.2 \pm 0.7) \times 10^4$ M$^{-1}$ (Supplementary Figure 12) and $(9.0 \pm 1.2) \times 10^4$ M$^{-1}$ (Supplementary Figure 13), respectively. These results demonstrate that the inclusion of $\mathit{p}$-C$_6$H$_4$Cl$_2$ within the cavity of $[\mathbf{CBPQT} \subset \mathbf{1}]^{4(+\bullet)}$ can help to stabilize the box-in-box assembled complex and enhance the binding affinity of $[\mathbf{CBPQT} \subset \mathbf{1}]^{4(+\bullet)}$.

The formation of a three-component inclusion complex was confirmed in the solid-state by single-crystal XRD characterization of the superstructure of $[\mathit{p}\text{-}\mathbf{C_6H_4Cl_2} \subset \mathbf{CBPQT} \subset \mathbf{1}]\cdot 4PF_6$. Single crystals of this assembly were obtained by slow vapor diffusion of $^i$Pr$_2$O into a mixture of 1 equiv of $[\mathbf{CBPQT} \subset \mathbf{1}]^{4(+\bullet)}$ and 4 equiv of 1,4-dichlorobenzene in MeCN. The superstructure shows (Fig. 8 and Supplementary Figure 16) that 1,4-dichlorobenzene molecule is located inside the cavity of the $[\mathbf{CBPQT} \subset$

$\mathbf{1}]^{4(+\bullet)}$ complex to form the desired Russian doll-like $[\mathit{p}\text{-}\mathbf{C_6H_4Cl_2} \subset \mathbf{CBPQT} \subset \mathbf{1}]^{4(+\bullet)}$ complex. The co-conformation of the two cyclophane components within the three-component complex remains almost the same (Fig. 6) as in the complex $[\mathbf{CBPQT} \subset \mathbf{1}]^{4(+\bullet)}$ in the absence of an additional guest, with the three-component complex exhibiting a contact distance between BIPY$^{\bullet+}$ units of 3.05 Å and a torsion angle of 86°. The distance between 1,4-dichlorobenzene and the adjacent BIPY$^{\bullet+}$ units is 3.55 Å, an observation that indicates the presence of $[\pi \cdots \pi]$ interactions between 1,4-dichlorobenzene and $\mathbf{CBPQT}^{2(+\bullet)}$. The two Cl atoms of the guest are located at the "windows" between the two cyclophanes and exhibit short Cl$\cdots$H contact distances (2.97 and 3.03 Å) with the two $\mathit{ortho}$ H-atoms of the bis(phenylene)ethyne units. This observation indicates the existence of weak [C–H$\cdots$Cl] hydrogen bonds between the diphenylethyne units and dichlorobenzene guest, which providing additional stabilization of the Russian doll assembly. The successful formation of a three-component complex with 1,4-dichlorobenzene prompted us to explore other 1,4-substituted benzene derivatives—namely, 1,4-dibromobenzene, dimethyl terephthalate, 1,4-bis(allyloxy)benzene and bis(propargyl)-terephthalate—as guest molecules to bind inside the $[\mathbf{CBPQT} \subset \mathbf{1}]^{4(+\bullet)}$ complex. All these molecules were found to form Russian doll assemblies (Fig. 8, Supplementary Figure 17-21) in the solid-state that are similar to the three-component complex formed with 1,4-dichlorobenzene. The contact distances between the BIPY$^{+\bullet}$ units of two cyclophane components are nearly identical (3.04–3.05 Å, Fig. 8) for all of the Russian doll assemblies. The torsion angles, however, vary significantly for different guest molecules—86° for 1,4-dichlorobenzene, 78° for 1,4-dibromobenzene, 74° for 1,4-bis(allyloxy)benzene, 70° for 1,4-bis(propargyl)-terephthalate and 66° for dimethyl terephthalate—reflecting the steric influence of the 1,4-substituents on these benzene derivatives. In addition, the accommodation of different guest molecules significantly influences the packing properties (Supplementary Figure 17-21) of the complexes. It is noteworthy that $[\mathbf{1,4\text{-}bis(allyloxy)benzene} \subset \mathbf{CBPQT} \subset \mathbf{1}]^{4(+\bullet)}$ and $[\mathbf{1,4\text{-}bis(propargyl)\text{-}terephthalate} \subset \mathbf{CBPQT} \subset \mathbf{1}]^{4(+\bullet)}$ are ring-in-ring-type pseudo[3]rotaxanes with vinyl or alkynyl functional groups at the end of the dumbbells, making these assemblies potential precursors for preparing new mechanically interlocked molecules[45,50] and redox-controlled molecular machines[51].

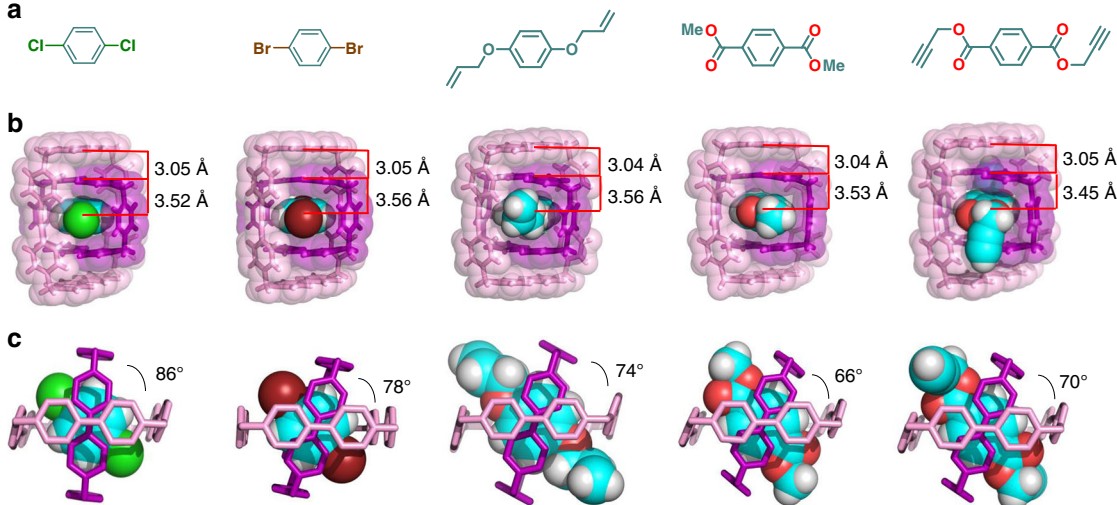

**Fig. 8** Solid-state superstructure of Russian doll assemblies with different guests inside the $[\mathbf{CBPQT} \subset \mathbf{1}]^{4(+\bullet)}$ complex. **a** Structural formulas of the guest molecules. **b** Perspective views depicted as tubular and space-filling representations. **c** Side-on views depicted as tubular representations. Hydrogen atoms are omitted for the sake of clarity

## Discussion

As detailed above, we have synthesized a tetracationic cyclophane that was designed to act as a size-complementary host for recognizing the smaller **CBPQT**$^{4+}$ cyclophane upon reduction of both cyclophanes to their diradical dicationic states. Radical-pairing interactions between the two cyclophanes drives the formation of a novel tetraradical tetracationic box-in-box complex [**CBPQT** $\subset$ **1**]$^{4(+\bullet)}$, which was investigated thoroughly in solution by spectroscopic techniques (UV–Vis–NIR, VT-EPR, VT-$^1$H-NMR) and cyclic voltammetry. An association constant of $K_a = (1.6 \pm 0.2) \times 10^4$ M$^{-1}$ was determined on the basis of UV–Vis–NIR titration experiments, and EPR measurements confirmed strong radical-pairing between the cyclophanes to produce a diamagnetic assembly. Remarkably, the radical-pairing interactions are strong enough to permit characterization of [**CBPQT** $\subset$ **1**]$^{4(+\bullet)}$ by $^1$H NMR spectroscopy—the first time this technique has been used for characterizing a supramolecular assembly held together by radical–radical interactions. Characterization by $^1$H NMR spectroscopy provided detailed information about the co-conformation of [**CBPQT** $\subset$ **1**]$^{4(+\bullet)}$ in solution, which was consistent with the solid-state superstructure that was determined by single-crystal XRD analysis.

The most notable feature of the box-in-box complex is its ability to bind various 1,4-disubstituted benzene derivatives inside the cavity of the smaller cyclophane to form hierarchical Russian doll assemblies, which is quite rare in supramolecular systems. Besides, the inclusion of third component guests within the cavity of the box-in-box complex can help to stabilize the assembled structures. These three-component assemblies were characterized in the solid-state by single-crystal XRD analysis, as well as in solution by $^1$H NMR spectroscopy. The formation of these hierarchical complexes highlights the box-in-box complex as an attractive motif for forming sophisticated supramolecular assemblies that could be used for designing complicated new mechanically interlocked structures and redox-controllable molecular machines. Thus, this research demonstrates the utility of precisely designing the dimensions of a host for binding a specific guest, such as a smaller cyclophane, that imbues the resulting host–guest complex with appealing properties and functionalities.

## Methods

**Materials**. Solvents used in experiments involving radicals were degassed using the freeze-pump-thaw method. The tetracationic cyclobis(paraquat-*p*-phenylene) tetrakis (hexafluorophosphate) (**CBPQT•4PF$_6$**) was prepared according to literature procedures.

**Synthesis of 3•2PF$_6$**. 4,4′-Bipyridine (4.0 g, 25.6 mmol) was dissolved in MeCN (60 mL), and the solution was heated to 90 °C. Compound **2** (1.3 g, 4.12 mmol) was added in three portions within 1 h. The solution was cooled to room temperature after refluxing for 24 h, and then TBACl (2.5 g, 9.0 mmol) was added. The precipitate was collected by filtration, washed twice with MeCN, and then dissolved in H$_2$O. Excess of NH$_4$PF$_6$ was added to the aqueous solution, and the precipitate was collected by filtration and washed successively with H$_2$O, MeOH, and Et$_2$O. After drying in air, **3•2PF$_6$** was obtained as white solid (3.09 g, 93%). $^1$H NMR (500 MHz, CD$_3$CN) δ 8.47–8.30, (m, 8 H), 8.32 (d, $J$ = 6.95 Hz, 4 H), 7.79 (d, $J$ = 6.25 Hz, 4 H), 7.65 (d, $J$ = 8.3 Hz, 4 H), 7.50 (d, $J$ = 8.2 Hz). $^{13}$C NMR (125 MHz, CD$_3$CN) δ 150.5, 146.0, 134.9, 133.1, 130.0, 128.2, 124.8, 90.6, 65.2.

**Synthesis of 1•4PF$_6$**. **3•2PF$_6$** (806 mg, 1.0 mmol), **2** (380 mg, 1.1 mmol) and TBAI (75 mg, 0.20 mmol) were dissolved in MeCN (600 mL), the solution was then heated to 90 °C and stirred for 5 days. The solution was cooled to room temperature, and TBACl (0.83 g, 3.0 mmol) was added to precipitate the solids. The precipitate was collected by filtration, washed twice with MeCN, and then dissolved in H$_2$O. Excess of NH$_4$PF$_6$ was added to the aqueous solution, the precipitate was collected by filtration and washed successively with H$_2$O and MeOH. The crude product was purified by column chromatography (SiO$_2$: 2% NH$_4$PF$_6$ in Me$_2$CO) to afford **1•4PF$_6$** as a yellow solid (525 mg, 42%). $^1$H NMR (500 MHz, CD$_3$CN) δ 8.99 (d, $J$ = 7.0 Hz, 4 H), 8.32 (d, $J$ = 7.0 Hz, 8 H), 8.98 (d, $J$ = 7.0 Hz, 8 H), 7.55 (dd, $J$ = 8.1 Hz, 12.6 Hz, 16 H), 5.81 (s, 8 H). $^{13}$C NMR (126 MHz, CD$_3$CN) δ 150.5, 146.0, 134.9, 133.1, 130.0, 128.2, 124.8, 90.6, 65.2. (HRMS-ESI). For **1•4PF$_6$**, Calcd for

C$_{52}$H$_{40}$F$_{24}$N$_4$P$_4$: $m/z$ = 1155.2178 [$M$–PF$_6$]$^+$, 505.1264 [$M$–2PF$_6$]$^{2+}$; found: 1155.2163 [$M$–PF$_6$]$^+$, 505.1264 [$M$–2PF$_6$]$^{2+}$.

**UV–Vis–NIR titration**. Stock solutions of the fully oxidized viologen derivatives **CBPQT•4PF$_6$**, **1•4PF$_6$** were prepared in an N$_2$ glovebox. The stock solutions were reduced over activated Zn dust for 10–15 min with stirring and then filtered to provide deep blue solutions of **CBPQT**$^{2(+\bullet)}$ or **1**$^{2(+\bullet)}$. Syringes were employed to measure and dilute the radical stock solutions to the desired concentrations prior to measurements. Spectra were recorded from 1300 to 400 nm in a sealed 2 mm path length cells during titrating **CBPQT**$^{2(+\bullet)}$ (0 to 10 equiv) into **1**$^{2(+\bullet)}$. Binding constant was obtained by fitting fit a 1:1 isotherm according to literature[47].

**EPR measurement**. Solutions of **CBPQT**$^{2(+\bullet)}$ and **1**$^{2(+\bullet)}$ were prepared in N$_2$ glovebox in the same way as was described for preparing the samples utilized for UV–Vis–NIR measurements. After mixing and/or diluting the radical samples to the desired concentration, 100 μL of each sample was transferred to a quartz EPR tube by syringe. The tubes were sealed with UV-cure resin under an N$_2$ atmosphere.

**Cyclic voltammetry measurement**. Samples for cyclic voltammetry were prepared using an electrolyte solution of 0.1 M Bu$_4$NPF$_6$ in MeCN that was sparged with Ar to remove O$_2$. The cyclic voltammograms presented in the main text of the manuscript were recorded under Ar or N$_2$ using a glassy carbon working electrode, a Pt wire or Pt mesh counter electrode, a silver wire quasi-reference electrode, and an internal standard of ferrocene.

**VT-NMR measurement**. The sample solutions were prepared in N$_2$ glovebox with a similar procedure as described for preparing the samples utilized for UV–Vis–NIR measurements. Variable-temperature $^1$H NMR spectra and low temperature $^1$H-$^1$H COSY NMR were recorded using an Agilent DD2 spectrometer with a 600 MHz working frequency for $^1$H nuclei.

**Crystallizations and X-ray analyses for all complexes**. For **1•4PF$_6$**: Single crystals were grown on the bench-top by slow vapor diffusion of $^i$Pr$_2$O into a 1.0 mM solution of **1•4PF$_6$** in MeCN over the course of a week. For [**CBPQT** $\subset$ **1**]•4PF$_6$ and all the five Russian doll assemblies: Excess of activated Zn dust was added to a mixture of **CBPQT•4PF$_6$** (1.1 mg, 1.0 μmol), **1•4PF$_6$** (1.3 mg, 1.0 μmol) together without additional guests ([**CBPQT** $\subset$ **1**]•4PF$_6$) or with 4 equiv of the corresponding 1,4-disubstituted benzene derivatives in an N$_2$ glovebox, and the mixtures were stirred for 20 min. After filtering, the purple solutions were kept under an atmosphere of $^i$Pr$_2$O at room temperature for a week to allow slow vapor diffusion to occur. The crystals, which appeared in the tubes, were selected and mounted using oil (Infineum V8512) on a glass fiber and transferred to the cold gas stream cooled by liquid N$_2$ on Bruker APEX-II CCDs with MX optics Mo-K$_\alpha$ or Cu-K$_\alpha$ radiation. The structures were solved by direct methods and refined subsequently using OLEX2 software. CCDC 1851540–185146 contain the supplementary crystallographic data for this article. These data can be obtained free of charge from the Cambridge Crystallographic Data Centre via www.ccdc.cam.ac.uk/data_request/cif.

## Data availability

All the data generated or analyzed during this study are included in this published article (and its supplementary information files) or available from the authors upon reasonable request. The crystallographic data in this study have been deposited in the Cambridge Structural Database under entry IDs CCDC 1851540–185146.

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

## Acknowledgements

This research is part of the Joint Center of Excellence in Integrated Nano-Systems (JCIN) at King Abdulaziz City of Science and Technology (KACST) and Northwestern University (NU). Computational investigations were supported by the U.S. National Science Foundation under grant no. EFRI-1332411 (W.A.G. and T.C.). We would like to thank both KACST and NU for their continued support of this research. Electron paramagnetic resonance studies were supported by the U.S. National Science Foundation under Grant No. CHE-1565925 (M.R.W.).

## Author contributions

K.C. conceived and designed the research, carried out most of experiments, analyzed the data and composed the manuscript. M.C.L. involved in the single-crystal growths, UV–Vis–NIR and CV measurements. J.-M.H. helped with the CV measurements. C.L.S. did the crystallographic analysis. M.R.W. and J.N. performed the EPR studies. W.A.G. and T.C. performed DFT calculations. M.C.L. and Z.L. did the first revision of the manuscript. Y.F. and S.V. provided some help with the synthesis. Y.S., D.S., and C.C. involved in the discussions and contributed to the manuscript preparation. J.F.S. directed the research. All authors discussed and commented on the manuscript.

## Additional information

**Competing interests:** The authors declare no competing interests.

