## [Peer Review File · Nature Communications]

Reviewers' comments:

Reviewer #1 (Remarks to the Author):

The manuscript by Cai, Stoddart and coworkers reported a box-in-box radical-pairing assembly which could further host smaller neutral guest molecules. The authors used "Russian Doll" to interpret the multi-layered structures, and I am very curious whether four-layered or five-layered structures could be achieved using such method. In general, this is a very challenging but interesting research. The following issues need to be addressed properly before consideration for publication.

1. Despite that the formula for calculating 1:1 binding constant is well known; the authors still need to specify the method they used for calculating the binding constants of host-guest complexes. Detailed information such as R² value of fitting equation should also be provided.
2. The authors claimed the radical-pairing interactions to be strong. I'm curious if host-guest complexes could be characterized by some soft ionization method such as ESI-MS or CSI-MS as previous report (Nat. Commun. 2017, 8, 634). Or is this radical system very air sensitive for mass spectrometry characterization?
3. The authors should provide detailed synthetic procedures and characterization data for all the compounds, such as 1•4PF₆⁻. If the compound has been previously reported, proper citation is needed.
4. The explanation for the failure of encapsulation of radical cationic dimethyl viologen (MV^{+•}) is somewhat unconvincing. According to the previous report (J. Am. Chem. Soc. 2017, 139, 3986) and the single-crystal X-ray diffraction analysis (larger BIPY^{+•} centroid-to-centroid distance (7.19 Å), as well as the encapsulation of a series of neutral molecules in this manuscript, window size shouldn't be a big issue. Is that possible the mixing order of the three components matters? Or the electronic environment inside the [CBPQTc1]₄^(+•) is different from that in the CBPQT₂^(+•)? DFT calculation may help.
5. The units of scan speed in Figure S11 should be mv/s.
6. The authors should provide VT-NMR spectrum for the host as well as guest molecule in order for comparison. Moreover, have the authors tried NMR for 1•2^(+•)? The chemical shift for 1•4PF₆ might be different from that of 1•2^(+•)

Reviewer #2 (Remarks to the Author):

A new tetracationic box-like macrocycle (1)₄⁺ has been prepared and characterized. This new box is large enough to accommodate the ubiquitous (CBQT)₄⁺ box via radical-radical pairing in a complete host-guest manner. The H-G adduct was characterized by optical spectroscopy (association constant), sc-XRD and remarkably, the radical-pairing interactions are strong enough to permit characterization of by ¹H NMR – the first time this technique has been used for characterizing a supramolecular assembly held together by radical-radical interactions. Since the H-G adduct, still contains a small cavity inside the assembly, complexation of small electron-rich molecules was explored. Five H-G-G assemblies were characterized by LT ¹H NMR and sc-XRD showing that the third component could be trapped inside the cavity formed by the other two. This is exceptional work that clearly demonstrates how two rigid 2D objects can be co-opted to create a 3D assembly into which a guest can be

sequestered. I recommend publication. Comment: Presumably, the small electron-rich guests are of the correct size to interact with (CBQT)⁴⁺ (or a radical version) only. It would be interesting to know to what degree (quantitatively) the binding increases when the third larger outside component is added to the assembly. This would provide a baseline for the concept.

Reviewer #3 (Remarks to the Author):

Cai et al present the synthesis and characterization of a new box-in-box cyclophane assembly, formed by radical-radical pairing interactions. The new assembly contains a sizable central void space, which allows for inclusion of additional guest molecules (1,4-disubstituted benzene derivatives). Characterization of the multi-component assemblies is presented in both solution and solid state. Future directions based on potential cross-linking of the guest molecules are proposed.

The creation of these hierarchical assemblies is impressive and novel, the experimental work is sound and well-described, and the paper is well-written. The solution-phase characterization approach presented here (VT NMR, in particular) is likely to be useful for others working in the field. Overall, I can recommend publication of this work in Nature Communications, though I have a few questions & suggestions.

One area that could benefit from additional study is a comparison of the temperature-dependent behavior in the solid state versus solution. In the manuscript, the authors state: "Upon heating from -20 °C to 80 °C, the EPR signal of the mixture gradually increased in intensity, along with an increase in the appearance of hyperfine structure. This observation is ascribed to the temperature induced dissociation of the box-in-box complex into its individual radical cyclophane components." This explanation is very reasonable, but it makes me curious about the thermal dependence of radical pairing in the solid state, where the assembly cannot readily dissociate. An easy starting point would be to repeat the VT EPR experiment using a crystalline sample. Magnetism experiments (e.g. SQUID on the crystalline sample versus VT Evans NMR on the solution) could also be interesting and informative. A related question that comes to mind is whether the inclusion of the 3rd guest component affects the interactions between the radicals in the assembly, since a continuous 5-layer pi-stacked assembly is now formed. Consideration of these questions could increase the impact of this study.

Response Letter to Reviewers' Comments

Reviewer 1

1. *Despite that the formula for calculating 1:1 binding constant is well known; the authors still need to specify the method they used for calculating the binding constants of host-guest complexes. Detailed information such as R^2 value of equation should also be provided.*

Reply

We thank this reviewer for these helpful comments. The details of 1:1 fitting method has already been reported in our previous paper (Ref. 47), and this paper has been cited in the titration part of our manuscript. The R^2 value of fitting equations have now been added to the *Supplementary Information* (Supplementary Figure S9-S13).

2. *The authors claimed the radical-pairing interactions to be strong. I'm curious if host-guest complexes could be characterized by some soft ionization method such as ESI-MS or CSI-MS as previous report (Nat. Commun. 2017, 8, 634). Or is this radical system very air sensitive for mass spectrometry characterization?*

Reply

We thank this reviewer for raising these questions. We tried ESI-MS measurement for the both box-in-box complex [CBPQTc1]•4PF₆ and the Russian doll complex [p-C₆H₄Cl₂cCBPQTc1]•4PF₆. We failed, however, to detect the desired signals for either of these complexes. Instead, the major signals comes from individual radical cationic cyclophanes or their oxidized derivatives. We ascribed this failure to the air-sensitive nature of the radical complexes. Since the radicals are easily oxidized during the measurement, and once the tetradical tetracationic complex is oxidized—even partially oxidized—the binding affinity of the complexes will be significantly weakened, leading to the dissociation of the complexes.

3. *The authors should provide detailed synthetic procedures and characterization data for all the compounds, such as 1•4PF₆. If the compound has been previously reported, proper citation is needed.*

Reply

The synthetic procedures and characterization data was already included in the manuscript (see Method part).

4. *The explanation for the failure of encapsulation of radical cationic dimethyl viologen (MV+•) is somewhat unconvincing. According to the previous report (J. Am. Chem. Soc. 2017,139, 3986) and the single-crystal X-ray diffraction analysis (larger BIPY+• centroid-to-centroid distance (7.19 Å), as well as the encapsulation of a serious of neutral molecules in this manuscript, window size*

shouldn't be a big issue. Is that possible the mixing order of the three components matters? Or the electronic environment inside the [CBPQT \subset 1] $^{4(+\bullet)}$ is different from that in the CBPQT $^{2(+\bullet)}$? DFT calculation may help.

Reply

We thank this reviewer for raising this question.

We agree that electronic environment inside the box-in-box complex should be quite different from that of CBPQT $^{2(+\bullet)}$. As the reviewer mentioned, the BIPY $^{+\bullet}$ centroid-to-centroid distance of the complex (7.19 Å) was enlarged about 0.29 Å compared with free CBPQT $^{2(+\bullet)}$ (6.90 Å), a situation which is unfavorable for the encapsulation of MV $^{+\bullet}$. The limited window size of the box-in-box complex should, however, also be an important factor in preventing the binding of MV $^{+\bullet}$. From the crystal structure of the box-in-box complex, we can observe that CBPQT $^{2(+\bullet)}$ and $\mathbf{1}^{2(+\bullet)}$ are bound together in an almost perpendicular arrangement, leaving the open window of CBPQT $^{2(+\bullet)}$ shielded by the diphenylethyne units of $\mathbf{1}^{2(+\bullet)}$, and therefore the open window of the whole [CBPQT \subset 1] $^{4(+\bullet)}$ complex is significantly diminished compared with that of free CBPQT $^{2(+\bullet)}$. Actually, the distance between the *p*-xylylene units of CBPQT $^{2(+\bullet)}$ and diphenylethyne units of $\mathbf{1}^{2(+\bullet)}$ is merely 5.2 Å, which is too small for a MV $^{+\bullet}$ guest to thread through. In addition, we also tried some other guests with more than one benzene ring such as anthracene and 2,6-dihydroxynaphthalene and we failed to obtain any evidence of the formation of Russian doll complexes in the case of these guests. Therefore, we ascribed the failure of encapsulation of MV $^{+\bullet}$ to the limited window size provided by the [CBPQT \subset 1] $^{4(+\bullet)}$ complex.

5. *The units of scan speed in Figure S11 should be mv/s.*

Reply

We thank this reviewer for noticing this point. The scan speed unit (V/s) in Figure S11 is correct, since we varied the scan rate from normal (200 mV/s) to ultrafast (100 V/s) to investigate the kinetics associated with the redox process. Such a scan-rate-varied cyclic voltammogram technique has been reported in some previous papers published by our group (see Ref. 44 and Ref. 47).

6. *The authors should provide VT-NMR spectrum for the host as well as guest molecule in order for comparison. Moreover, have the authors tried NMR for $\mathbf{1}^{2(+\bullet)}$? The chemical shift for $\mathbf{1}\cdot\text{4PF6}$ might be different from that of $\mathbf{1}^{2(+\bullet)}$*

We thank this reviewer for raising this question. We recorded the VT NMR spectra for both $\mathbf{1}^{2(+\bullet)}$ and CBPQT $^{2(+\bullet)}$ in CD₃CN from 25 °C to -40 °C, but no any NMR signals were observed even at -40 °C. This observation is understandable since the single electrons of the radical cationic viologens in $\mathbf{1}^{2(+\bullet)}$ and CBPQT $^{2(+\bullet)}$ are unpaired, and so they are NMR-silent even at low temperature.

Reviewer 2

Presumably, the small electron-rich guests are of the correct size to interact with $(\text{CBPQT})^{4+}$ (or a radical version) only. It would be interesting to know to what degree (quantitatively) the binding increases when the third larger outside component is added to the assembly. This would provide a baseline for the concept.

Reply

We thank this reviewer for raising these very useful and important points.

We did some additional UV-Vis-NIR titration experiments (see below) with $\mathbf{1}^{2(+)}$ and $\text{CBPQT}^{2(+)}$, using the same protocol described in the manuscript for the measurement of the binding affinity of $[\text{CBPQT}\subset\mathbf{1}]^{4(+)}$. In the initial solution of $\mathbf{1}^{2(+)}$, however, we now added some equivalents of the small guest $p\text{-C}_6\text{H}_4\text{Cl}_2$, then $\text{CBPQT}^{2(+)}$ was titrated into the mixture solution and the “apparent” binding constant of $[\text{CBPQT}\subset\mathbf{1}]^{4(+)}$ complex was measured using the same 1:1 fitting isotherm as before. Interestingly, the existence of the $p\text{-C}_6\text{H}_4\text{Cl}_2$ guest indeed increased the binding affinity between $\mathbf{1}^{2(+)}$ and $\text{CBPQT}^{2(+)}$, and the more equivalents of $p\text{-C}_6\text{H}_4\text{Cl}_2$ that were added, the more the binding affinity increased. For instance, when 2.0 equiv $p\text{-C}_6\text{H}_4\text{Cl}_2$ was present, the “apparent” binding constant is $(3.8 \pm 0.3) \times 10^4 \text{ M}^{-1}$, which is 1.3 times higher than that of 0 equiv $p\text{-C}_6\text{H}_4\text{Cl}_2$; and when the equivalent of DCB was 6.7 equiv and 20 equiv, the “apparent” binding constants are measured to be $(6.2 \pm 0.7) \times 10^4 \text{ M}^{-1}$ and $(9.0 \pm 1.2) \times 10^4 \text{ M}^{-1}$, respectively. Besides, the NIR absorption peak of the radical complex was 11 nm redshifted (921 nm), compared with that of $[\text{CBPQT}\subset\mathbf{1}]^{4(+)}$, indicating that the encapsulation of $p\text{-C}_6\text{H}_4\text{Cl}_2$ leads to some electronic delocalization within the Russian doll complex. Therefore, binding $p\text{-C}_6\text{H}_4\text{Cl}_2$ into the cavity of $[\text{CBPQT}\subset\mathbf{1}]^{4(+)}$ complex can indeed help to stabilize the box-in-box complex.

Figure S11. a) Stacked UV-Vis-NIR spectra obtained by titrating $\text{CBPQT}^{2(+)}$ (0 to 8 equiv) into a 1:2 mixture solution of $\mathbf{1}^{2(+)}$ (0.15 mM) and 1,4-dichlorobenzene (0.30 mM); b) Binding isotherm simulation. Solvent: MeCN. Optical length: 2 mm.

Figure S12. a) Stacked UV-Vis-NIR spectra obtained by titrating $\text{CBPQT}^{2(+)}$ (0 to 8 equiv) into a 1:6.7 mixture solution of $\mathbf{1}^{2(+)}$ (0.15 mM) and 1,4-dichlorobenzene (1.0 mM); b) Binding isotherm simulation. Solvent: MeCN. Optical length: 2 mm.

Figure S13. a) Stacked UV-Vis-NIR spectra obtained by titrating $\text{CBPQT}^{2(+)}$ (0 to 8 equiv) into a 1:20 mixture solution of $\mathbf{1}^{2(+)}$ (0.15 mM) and 1,4-dichlorobenzene (3.0 mM); b) Binding isotherm simulation. Solvent: MeCN. Optical length: 2 mm.

We have added these titration results to the *Supplementary Information* (Figure S11-S13). And we also added a paragraph in the manuscript to describe these new results:

“One matter that is worth investigating is whether the inclusion of the third component guest ($p\text{-C}_6\text{H}_4\text{Cl}_2$) will affect the binding affinity between $\text{CBPQT}^{2(+)}$

and $\mathbf{1}^{2(+)}$. Thus, we repeated the UV-Vis-NIR titration experiments with $p\text{-C}_6\text{H}_4\text{Cl}_2$ presented in the solution. By titrating $\mathbf{CBPQT}^{2(+)}$ into the solution mixture of $\mathbf{1}^{2(+)}$ and a number of equivalents of $p\text{-C}_6\text{H}_4\text{Cl}_2$, similar 1:1 fitting isotherms can be applied to the titration data, and “apparent” binding constants were obtained. Notably, the existence of $p\text{-C}_6\text{H}_4\text{Cl}_2$ guests did, indeed, increase the binding affinity between $\mathbf{1}^{2(+)}$ and $\mathbf{CBPQT}^{2(+)}$; the more equivalents of $p\text{-C}_6\text{H}_4\text{Cl}_2$ that were added, the more the binding affinity increased. For instance, when 2.0 eq. $p\text{-C}_6\text{H}_4\text{Cl}_2$ was present, the “apparent” binding constant was found to be $(3.8 \pm 0.3) \times 10^4 \text{ M}^{-1}$ (Supplementary Figure S11), which is 1.3 times higher than that of the box-in-box complex without $p\text{-C}_6\text{H}_4\text{Cl}_2$; and when the equivalents of $p\text{-C}_6\text{H}_4\text{Cl}_2$ were 6.7 and 20, the “apparent” binding constant increased to $(6.2 \pm 0.7) \times 10^4 \text{ M}^{-1}$ (Supplementary Figure S12) and $(9.0 \pm 1.2) \times 10^4 \text{ M}^{-1}$ (Supplementary Figure S13), respectively. These results demonstrate that the inclusion of $p\text{-C}_6\text{H}_4\text{Cl}_2$ within the cavity of $[\mathbf{CBPQT}\subset\mathbf{1}]^{4(+)}$ can help to stabilize the box-in-box assembled complex and enhance the binding affinity of $[\mathbf{CBPQT}\subset\mathbf{1}]^{4(+)}$.

Reviewer 3

One area that could benefit from additional study is a comparison of the temperature-dependent behavior in the solid state versus solution. In the manuscript, the authors state: "Upon heating from -20 °C to 80 °C, the EPR signal of the mixture gradually increased in intensity, along with an increase in the appearance of hyperfine structure. This observation is ascribed to the temperature induced dissociation of the box-in-box complex into its individual radical cyclophane components." This explanation is very reasonable, but it makes me curious about the thermal dependence of radical pairing in the solid state, where the assembly cannot readily dissociate. An easy starting point would be to repeat the VT EPR experiment using a crystalline sample. Magnetism experiments (e.g. SQUID on the crystalline sample versus VT Evans NMR on the solution) could also be interesting and informative.

Reply

We thank this reviewer for raising these useful and points.

We recorded the EPR spectrum (see below) for 1 mg powder crystalline sample of $[\mathbf{CBPQT}\subset\mathbf{1}]^{4(+)}$. There is a radical signal of low signal-to-noise ratio in the EPR spectrum. Additionally, as expected from the ensemble of orientations in the powder sample, its EPR spectrum loses all hyperfine resolution and is naturally narrower than that of the solvated sample. Regardless of these significant differences, the VT-EPR experiment was repeated on the powdered crystalline sample from room up to higher temperatures. In general, increased noise in the VT-EPR experiment is caused by distortion of the quality (Q) of the resonator on account of the increased flow of evaporated liquid nitrogen (used to maintain the temperature by cooling the heating

element) through the resonator at higher temperatures. Slight increases in the noise are more significant at this lower ratio of signal to noise, and so the VT-EPR was only measured up to 323 K.

As the reviewer suggests, the key difference in the crystalline sample is that the assembly cannot readily dissociate. As a result, the VT-EPR spectrum of $[\text{CBPQT}\text{C}1]^{4(+)}$ from 293 to 323 K does not appreciably change except for the signal-to-noise (as explained above). This observation strengthens the argument that the thermal dependence of the EPR signal observed for $[\text{CBPQT}\text{C}1]^{4(+)}$ in MeCN is a result of the dissociation of the box-in-box complex into its individual radical cyclophane components.

Figure S23. VT-EPR Spectra of crystalline sample (1mg) of $[\text{CBPQT}\text{C}1]\cdot 4\text{PF}_6$ from 293K to 323K.

A related question that comes to mind is whether the inclusion of the 3rd guest component affects the interactions between the radicals in the assembly, since a continuous 5-layer pi-stacked assembly is now formed. Consideration of these questions could increase the impact of this study.

Reply

We thank this reviewer for these helpful and insightful comments.

The fact that the lengthening of the π -stacking upon inclusion of the third guest component may affect the interactions of the radicals in the assembly is worthy of investigation. We measured the EPR spectra (see below) of 0.50 mM $[p\text{-C}_6\text{H}_4\text{Cl}_2\text{C}\text{CBPQT}\text{C}1]^{4(+)}$ in MeCN at 293 K. It exhibits more pronounced hyperfine than that for $[\text{CBPQT}\text{C}1]^{4(+)}$ at 293 K. This observation is perhaps the opposite of what one might expect from increased π -stacking and potentially increased electronic delocalization.

With regard to molecular orbital theory, the slight change in the expression of the hyperfine structure could be the result of the complex nodal or *anti*-nodal structure of the SOMO of the Russian doll complex. From an electrostatic perspective, the polarizable Cl on *p*-C₆H₄Cl₂ may be shunting or drawing electron density to have a stronger hyperfine interaction with H¹ or N¹⁵ nuclei in the three-component complex. Ultimately, this slight, but unexpected change in the EPR spectrum of [p-C₆H₄Cl₂⊂CBPQT⊂1]^{4(+)•} is good evidence of the potential to tune finely the electronic structure of the box-in-box complex with the choice of appropriate guests.

Figure S24. EPR spectrum of [p-C₆H₄Cl₂⊂CBPQT⊂1]•4PF₆ (red line) and [CBPQT⊂1]•4PF₆ (blue line) at 293K in MeCN (0.50 mM for each).

We have added these new results to the *Supplementary Information* (Figure S23-S24).

REVIEWERS' COMMENTS:

Reviewer #1 (Remarks to the Author):

The authors have addressed all the questions and concerns of reviewer's. I have no further problem for the publication.

Reviewer #2 (Remarks to the Author):

The authors have done an excellent job addressing my concerns. The new data enhances the article. I recommend publication.

Reviewer #3 (Remarks to the Author):

The revised manuscript by Cai et al adequately addresses my comments from the original submission; I am also satisfied with the authors' responses to the comments of the other two reviewers. I recommend publication of the revised manuscript.

I still think that an improved understanding of the electronic structure of the three-component assemblies will be valuable for future work (as potentially modulated by the choice of 'inner' guest, which was suggested in the authors' response letter). But I think that this work is appropriate for a later study, and is not necessary for this manuscript.